# A Review of Psychological Stress among Students and Its Assessment Using Salivary Biomarkers

**DOI:** 10.3390/bs12100400

**Published:** 2022-10-18

**Authors:** Bruno Špiljak, Maja Vilibić, Ana Glavina, Marija Crnković, Ana Šešerko, Liborija Lugović-Mihić

**Affiliations:** 1School of Dental Medicine, University of Zagreb, 10000 Zagreb, Croatia; 2Department of Dermatovenerology, University Hospital Center Sestre Milosrdnice, 10000 Zagreb, Croatia; 3Department of Psychiatry, University Hospital Center Sestre Milosrdnice, 10000 Zagreb, Croatia; 4School of Medicine, Catholic University of Croatia, 10000 Zagreb, Croatia; 5Department of Oral Medicine and Periodontology, Dental Clinic Split, School of Medicine, University of Split, 21000 Split, Croatia; 6Center for Child and Youth Protection, 10000 Zagreb, Croatia; 7Department of Gynecology, University Hospital Center Zagreb, 10000 Zagreb, Croatia

**Keywords:** stress, psychological factors, psychological disturbances, students, saliva, biomarkers, cortisol, amylase, HPA axis

## Abstract

Numerous psychoneuroimmune factors participate in complex bodily reactions to psychological stress, and some of them can be easily and non-invasively measured in saliva (cortisol, alpha-amylase, proinflammatory cytokines). Cortisol plays a crucial role in the stress response; thus, stressful events (academic examinations, cardiac surgery, dental procedures) are accompanied by an increase in cortisol levels. (A correlation between cortisol blood levels and salivary values has already been confirmed, particularly during stress). Academic stress is defined as everyday stress among students that has an impact on aspects of their psychological and physiological well-being. For example, exams are considered one of the most acute stressful experiences for students. The strength of the association between academic self-efficacy, psychological stress, and anxiety depends on a variety of factors: the type of academic challenge (e.g., oral exam), the presence of an audience, etc. Higher stress levels were predominantly recorded among younger students, primarily regarding their academic tasks and concerns (grades, exams, competing with peers for grades, fear of failing the academic year, etc.). The measurement of stress levels during academic stress can improve our understanding of the character and influence of stressful events in populations of students, preventing adverse reactions to long-term stress, such as a decreased immune response and increased anxiety.

## 1. Introduction

Due to the influence of various stressful stimuli, psychological stress alters the homeostasis of the organism. Consequently, the organism reacts, and the sympathetic-adrenal-medullary (SAM) system and the hypothalamic-pituitary-adrenal (HPA) axis are activated, producing and releasing specific hormones [1,2]. Therefore, the psychological response to emotional stress also modulates the functions of the immune system, the autonomic nervous system (ANS), levels of hypothalamic and pituitary hormones, neuropeptides, cytokines, and other factors involved in this network [3,4]. 

The stress response results in changes at the molecular level of the whole body (Figure 1) [5,6,7,8,9]. Thus, short-term and long-term effects of stress are associated with changes/alterations in HPA axis functioning, which alters glucocorticoid levels and may influence different health outcomes [10]. The stress response includes the connection between the central nervous system (CNS) and the immune system, with bidirectional connections. In addition, adaptation to stress is a very important mechanism in the body’s response to stress. Thus, the effectiveness of the reaction to stress depends on the type, intensity, and duration of stress as well as the characteristics of the person [11]. 

The aim of this narrative review is to present current data on psychological stress among graduate students and the possibilities for its assessment using salivary biomarkers. Research findings are based on articles published in English, available through the PubMed database and other prominent sources of research literature. 

## 2. Physiology of Acute and Chronic Stress Reactions and the HPA Axis

A stress reaction implies greater activity of the HPA axis. The organism’s response to acute stress involves a complex process/network which is mediated by the HPA axis and involves changes in psychological and social factors. In addition to acute stress, chronic psychological stress also activates the HPA axis, which causes elevated glucocorticoid levels [12]. It is well-known that this release of stress-associated glucocorticoids can impair memory and cognitive functions [1,13,14,15]. Chronic stress can also weaken the immune response (as confirmed by the determination of antibody responses to vaccines) and can cause or contribute to different diseases such as cardiovascular, endocrine, gastrointestinal diseases, and others [12]. The research literature indicates that acute stress can also change levels of different immune factors, but by increasing them [16]. Increased cortisol levels lead to hyperactivity of the HPA axis, while disorders of the HPA axis present an increased risk for developing various diseases and conditions [17,18]. However, research on the impact of naturally stressful events on diseases has given contradictory results about the increased risk of developing certain diseases [10]. 

The peripheral and central nervous systems are important factors in a complex network of reactions involved in the body’s response to stress [11]. The brain receives and processes various neurosensory impulses (cortical, limbic, visual, somatosensory, nociceptive, visceral, etc.), including signals coming from the blood (hormones, cytokines, mediators). During psychological stress, the crucial role belongs to the sympathetic nervous system, mostly to the ANS, which controls the functions of the internal organs via the sympathetic and parasympathetic nervous systems [11]. The acute stress response includes, firstly, the registration of the stressful stimulus, which is transmitted from the cerebral cortex to the nucleus of the brainstem locus coeruleus/norepinephrine (LC/NA) and pons, whose neurons stimulate (via receptors) the release of catecholamines from the adrenal medulla directly into circulation. This release of catecholamines affects the body by making an individual more alert and cautious, activating defensive behavior patterns, usually with higher aggression, stimulating the cardiovascular and respiratory systems and inhibiting the gastrointestinal system. Consequently, the increase in catecholamine levels also results in increased levels of plasma glucose and fatty acids. These reactions are crucial for survival and protection of the organism during stressful periods [11]. 

Stressors stimulate the hypothalamus to secrete corticotropin-releasing hormone (CRH) and arginine vasopressin (AVP), which leads to the release of adrenocorticotropic hormone (ACTH) from the anterior pituitary lobe. This is followed by activation of adrenal gland cells, which produce and release glucocorticoids. Additionally, HPA axis activation is crucial for the body’s catabolic processes and for supplying the organism with energy. Increased plasma cortisol concentration stimulates the liver to perform the process of gluconeogenesis and causes insulin resistance in peripheral tissues [19]. Consequently, glucocorticoids are crucial in controlling the duration of the stress response. Activation of the stress system causes clinical manifestations that include physiological reactions (oxygen and nutrients are directed to the CNS and other body systems) and behavioral reactions (e.g., increased excitement, vigilance, caution, focus, euphoria, or dysphoria, etc.) [11]. Clearly, multiple systems are involved in the adaptation of the body to new circumstances, and numerous processes (e.g., lipolysis and gluconeogenesis) take place to supply the body with energy at such times. At the same time, the activities of other systems (gastrointestinal, reproductive, and immune systems) are inhibited [11]. 

Successful adaptation to stress activates the body’s mechanisms that control and inhibit stress reactions—in other words, they prevent an excessive response to stress. Without these mechanisms in place, the stress response would be too intense and prolonged, going beyond successful adjustment, thus potentially causing a pathological condition. There is a constant, dynamic balance between the stimulatory and inhibitory mechanisms of a stress reaction, but the mechanisms that inhibit stress reactions prevail during the recovery process. A proper response to stress involves timely calming or neutralization of stress effects. Catabolic processes predominate after a stressful situation, i.e., energy mobilization occurs due to the effects of stress hormones (catecholamines, cortisol, glucagon) [20]. During rest and recovery, anabolic processes predominate due to the influence of growth hormone and gonadal steroids important for healing and growth. Thus, to establish the homeostasis that each cell tries to achieve, this balance between the anabolic and catabolic processes is needed [11]. 

Psychological stress is associated with various diseases and conditions, including cardiovascular diseases (e.g., hypertension), diabetes, gastrointestinal disorders, increased susceptibility to infections, autoimmune disorders, malignant diseases, and others [21]. In addition, high levels of stress have an impact on mental health and can lead to drug abuse, reduced work efficiency, absenteeism, and other behaviors associated with poor mental or overall health [2,22,23]. Literature data illustrates how medical interventions (pharmacological, psychotherapeutic) try to reduce the impact of cortisol in response to stressful stimuli [10].

It is important to mention, here, that gender influences the stress response—psychoendocrinological studies have reported lower stress responses in women than in men [24,25,26]. In addition, a subjectively higher perception of stress in women, and their typical cognitive styles, likely contribute to their higher risk of developing psychological disorders and anxiety disorders [27]. Gender’s significant impact on the immune system has been confirmed by several studies that analyzed the acute stress response. Notably, the menstrual cycle affects endocrine and immune variations by altering the level of circulating cytokines or growth factors [28]. A blunting impact on the stress response may be caused by fluctuations of gonadal steroids during the menstrual cycle and may also be related to oral contraceptive use [27,29]. Thus, Helbig et al. [27] showed a stronger cortisol response in women in the luteal phase than in those in the follicular phase or in those taking oral contraceptives. Additionally, salivary secretion differs between the sexes and may be associated with variations in the secretion of gonadal steroids and ANS regulation of salivary glands [30]. 

## 3. Immunity during Stress: The Inflammatory Reaction and Stress

The impact that stressful events can have on the immune system is a significant factor that affects the body’s ability to defend itself against diseases. Stress reduces the activity of natural killer (NK) cells and the proliferation of lymphocytes to specific viral antigens or nonspecific mitogens of B- or T-lymphocytes [31,32]. Stress is also associated with lower values of secretory immunoglobulin A (s-IgA) [33]. 

The relationship between psychological stress and immune parameters (e.g., immune cells, enzymes, cytokines) could be important and so are analyzed using various diagnostic tools [28]. There has been a lot of research in this area because a better understanding of the relationship could reveal useful data on how psychological stress affects the body. There are a few important mediators/cytokines linking the CNS and the immune system: TNF-α, IL-1β, IFN-γ, nerve growth factor (NGF)-β, which can activate the HPA axis [34,35]. Additionally, cytokines IL-1β, TNF-α, and IL-6 are crucial and reliable peripheral biomarkers associated with depression [36]. In addition, studies have revealed that macrophage-derived cytokine migration inhibitory factor (MIF) plays an important role in stress-immune network/interactions. Thus, MIF has proinflammatory, metabolic and angiogenic effects and participates in the pathogenesis of many diseases/disorders (inflammatory diseases, autoimmune diseases, atherosclerosis) and physiologic processes (e.g., wound healing), oncogenesis, metabolic disorders, etc. Animal studies have shown that hypothalamic CRH stimulates the secretion of MIF and thus influences NK cell activities important for inflammatory processes [37]. A particularly meaningful immune factor is IL-1β, especially for immune signaling and induction of the inflammatory process mediated by NK cells; it is linked to inflammatory tissue destruction [24]. 

The results of studies on the influence of stress on serum IL-1β are contradictory, i.e., some reports found that IL-1β levels during a stressful period had increased while others found that they had decreased [19,38,39]. Most studies in this area assessed the stress response and confirmed that IL-1β responds to antigen stimulation (in vivo and in vitro), e.g., there is a stress-associated increase [24,40,41,42,43]. However, a few researchers, it should be noted, found no effect on IL-1β [44,45,46] and some even observed a decrease [47].

## 4. Salivary Biomarkers for Stress Assessment

Different types of stress/stressful events are associated with endogenous oxidative and inflammatory stress, key sources of different biomarkers. Inflammatory reactions generate free radicals that generate other inflammatory mediators, such as IL-6, TNF-α, IL-10 or IL-1β [45]. Such circumstances lead to the organism’s greater susceptibility to infection [33]. Therefore, inflammation has been identified as a major etiological factor of various diseases/conditions (e.g., myocardial infarction, cardiovascular diseases, metabolic syndrome, diabetes, cancer, rheumatoid arthritis, neuropsychiatric disorders, etc.) [4,48,49,50,51]. 

The complexity of stress mechanisms makes acute stress measurement difficult to quantify and interpret. Because of this, the use of many subjective and objective diagnostic methods has been suggested, including salivary stress biomarkers (objective) [52]. Indicators of endocrine stress-associated changes include classical stress biomarkers, such as levels of hormones like cortisol and epinephrine [53]. Along with the ANS and the immune system, there are other HPA axis-associated factors involved in the stress response that present potential stress biomarkers (cortisol, alpha-amylase, and proinflammatory cytokines) (Figure 2) [4,9]. According to literature data, the most valuable salivary markers of stress, potentially, are cortisol, salivary alpha-amylase (sAA), chromogranin A and lysozyme [54]. However, stress hormones (e.g., glucocorticoids), along with cytokines, could also be very useful since they mediate various conditions, including post-traumatic stress disorder (PTSD). They represent important homeostatic regulators but are also promising functional stress biomarkers and can therefore help in the development of new therapeutics [55].

Stress induces several physiological and endocrine changes, disorders of functional parameters and changes in biochemical indicators [56]. This cascade affects the rapidly responding vital systems (primarily cardiovascular and CNS), but also other organ systems (hepatobiliary, pancreatic). Multisystem changes lead to disturbances of various stress-related biomarkers and factors which participate in this complex process [57]. When analyzing stress levels of persons without a psychiatric diagnosis, symptoms of anxiety and depression were associated with blunted/exaggerated cortisol responses to, and recovery from, stress, which may indicate an increased risk for unhealthy HPA axis dysregulation, allostatic load and disease [58]. Additionally, when analyzing HPA axis activation and cortisol levels of patients suffering from autoimmune diseases (systemic lupus erythematosus, Sjögren’s syndrome, and systemic sclerosis), changes in cortisol levels during the day were recorded (a higher area under the curve for cortisol levels was seen) [59]. Exaggerated stress-reactions have generally been viewed as maladaptive, with some evidence showing that individuals with exaggerated cardiovascular stress responses are at increased risk for developing cardiovascular disease due to various manifestations: hypertension, systemic atherosclerosis, coronary artery calcification, left ventricular hypertrophy, increased cardiovascular disease mortality, etc. [60]. Likewise, exaggerated cortisol reactivity was associated with coronary artery calcification and increased hypertension and cardiovascular disease risk. In healthy, older participants without a history/signs of coronary heart disease, there was an association between heightened cortisol reactivity and higher coronary artery calcification, which confirms that heightened HPA activity is a risk factor for coronary heart disease [61]. In addition, concerning stress and salivary cortisol in oral diseases, patients with oral lichen planus significantly exhibited higher stress, depression and anxiety scores compared to controls, with frequently increased cortisol levels (among 56.6% patients) and a positive correlation between psychological factors and salivary cortisol levels [62]. Stress-induced disorders of physiological, endocrine, immune, and metabolic functions lead to a complex cascade of oxidative, inflammatory, genomic, and proteomic responses responsible for excessive production of stress biomarkers [63,64]. Stress-induced reactions also include the participation of pro-inflammatory cytokines and oxidants of one generation that lead to/induce the formation of the second generation, which is followed by other factors such as reactive oxygen species (ROS) and inflammatory mediators [65]. Finally, complex disorders of the homeostatic mechanism cause alterations in protective defense mechanisms and result in different levels and types of stress responses, activating a genomic and proteomic response that expresses genes translated into proteins. It should be mentioned here that some metabolic, inflammatory, oxidative, genomic and proteomic factors can be used as stress biomarkers [63,64]. Thus, some biomarkers (cortisol, alpha-amylase, proinflammatory cytokines) have been established as stress biomarkers that reflect both SAM and HPA activity. Among various factors that belong to the neuroendocrine axis, cortisol plays a crucial role in the stress response [4]. Although new stress biomarkers are being investigated, recent studies have focused on cortisol of various origins [66]. It is well-known that increased cortisol levels are concomitant parameters during stressful events (academic examinations, cardiac surgery, dental procedures, etc.). According to the literature, there is a correlation between blood cortisol levels and salivary cortisol values, observed particularly during stress [67]. The HPA-axis regulates the secretion of cortisol, while the SAM regulates catecholamine secretion. After stress-induced activation of the HPA system, cortisol is secreted into the blood; thus, measuring serum and salivary cortisol levels can be a reliable indicator of stress. Salivary cortisol is mainly found in its unbound (free) form, and it accounts for about 70% of the total unbound cortisol in the organism [68]. Therefore, salivary cortisol is a useful biomarker of stressful conditions. However, the literature does not show uniform results regarding the association of stress and cortisol [54,69]. Another useful stress biomarker is salivary immunoglobulin A (s-IgA), which presents an antibody involved in the prevention of infectious diseases. Stress causes immune changes where s-IgA plays an important role as a biomarker of immune activation [70]. Salivary IgA levels are the first line of defense against various diseases and disorders such as upper respiratory tract disease, caries, and oral infections [16]. Previous studies have shown that acute stress causes activation of s-IgA, while chronic stress influences deleterious health consequences including decreased s-IgA levels [16,54,71]. Literature data shows that psychological stress is inversely related to IgA levels. Additionally, basal s-IgA levels are a potential indicator of health outcomes during stressful periods [16,54,72]. Recent studies have shown that an acute stress response causes a significant increase in salivary alpha-amylase (sAA) levels, and increased sAA concentrations have been reported as indicators and biomarkers of stressful situations [73]. According to the literature, salivary biomarkers are profoundly helpful in the diagnosis of various stress-related diseases including cancers, liver diseases, kidney diseases, neurological and cardiovascular diseases, psoriasis, systemic lupus erythematosus, rheumatoid arthritis [4,50]. Concerning potential influences on levels of salivary biomarkers, the impact of a higher body temperature and exercise on salivary biomarker levels indicates they significantly increase concentrations, i.e., significantly higher levels of cortisol, salivary alpha-amylase and total proteins [4,74]. Concerning s-IgA levels, much of the evidence indicated that mental imaging, relaxation, watching humorous videotapes, writing poetry, progressive muscle relaxation, and hypnosis are effective procedures for increasing s-IgA levels [16]. However, salivary biomarkers could be indicators of various biological processes (e.g., pathogenic or pharmacological responses) [4,53]. Concerning the use of biomarkers for measurements of stress levels among students (academic stress), there are various methods. Generally, questionnaires on stress are commonly used in practice for subjective data. The Scale for Assessing Academic Stress (SAAS), a 30-item self-reporting tool with “Yes” or “No” answers, measures students’ perceived stress [75]. The Perceived Stress Scale-10 (PSS-10) is also a frequently used scale to assess stress levels and evaluates the degree to which external demands appear to be higher than an individual’s perceived capability to handle the situation [76]. In contrast, objective indicators of stress are biomarkers analyzed primarily by blood or saliva. Since analysis of blood involves the invasive procedure of taking blood, salivary biomarkers provide a non-invasive, and thus advantageous, method of analysis. A few potential methods for analysis of salivary biomarkers exist: Enzyme-Linked Immuno Assays (ELISA), Radio Immuno Assays (RIA), as well as specific and sensitive techniques based on liquid chromatography coupled with tandem mass spectrometry (LC-MS/MS) [77,78]. The most commonly analyzed salivary biomarker is cortisol. It has been used for more than 30 years, including in studies on different populations, and it has been confirmed as a valid and sensitive method which exhibits parallelism, precision and accuracy [77]. Still, it should be taken into account that various factors can affect salivary cortisol results: sex hormone cycles, menstrual cycles, hormonal contraception usage, puberty, pregnancy, breastfeeding, menopause, etc. [68]. There are also other influences related to diagnostic procedures for salivary biomarkers; thus, although salivary cortisol concentration is not influenced by salivary flow rate and pH, this influence is possible during measurement of the salivary alpha-amylase activity [78]. In addition, there are some possible influences related to sampling procedures with saliva [79]. For example, procedures for collecting oral fluid (saliva) samples may differ, e.g., collecting may include non-stimulated and stimulated oral fluid samples [80]. This may influence the results, e.g., stimulation of the oral sample (saliva) allows for the collection of large sample volumes in a short time, limiting the variability of salivary pH [80].

## 5. Students and the Concept of Academic Stress

Academic stress is defined as everyday stress among students that has an impact on aspects of their psychological and physiological well-being [21]. Previous studies have noticed that younger students have higher stress levels than older students regarding their academic tasks and concerns (grades, exams, competing with peers for grades, fear of failing the academic year) [80]. Exams are considered one of the most acute stressful experiences for students [13]. 

Studies on stress reactions among students are common in the literature, and those that analyze the impact of academic stress on students’ reactions have provided useful data, as presented in Table 1, which is based on papers not older than 20 years [1,10,12,13,17,27,28,29,37,71,74,81,82,83,84]. According to a study by Bardi et al. [81], which monitored students during summer exams, students who obtained passing grades and had a better grade point average (GPA) also had a greater ability to remain calm during psychological stress and overall had higher levels of dehydroepiandrosterone (DHEA), a hormone that minimizes the negative effect of stress induced HPA axis activity. Other studies suggest that increased stress levels may be associated with an improved working memory and more pronounced emotional responses [81].

Previous studies show that the strength of the association between academic self-efficacy, anxiety, and psychological stress depends on a variety of factors—the type of academic challenge (e.g., oral exam), the presence of an audience, etc. Although the results of various studies differ, some studies have shown a significant association between academic stress and a decreased immune response on the one hand and certain immune aspects on the other [85]. The heterogeneity of the results is likely due to the fact that not only the impact of academic stress was observed and that some of the studies were not conducted on humans [33]. Many studies on academic stress have been conducted with medical and dental students, possibly because many researchers are doctors who work with students. Previous research notes that medical students experience stress for multiple reasons, including adaptation to the medical environment, exposure to human suffering and death, the imbalance between effort invested in their work and the rewards received combined with the anxiety of having numerous exams and other academic challenges, etc. According to literature data, medical students’ NK cell activities on the first day of final exams were lower than the month prior to exams [11,86,87]. In addition, Kiecolt-Glaser et al. [87] compared Epstein–Barr virus-triggered lymphocyte proliferation in medical students on the first day of the academic year and the final exam period with the month prior to exams and one week after returning from summer vacation. They noted reduced proliferation in the majority of the seropositive students during final exams [87]. Concerning dental students, research results show that clinical work or in-hospital obligations cause the greatest stress [80]. The stress experienced by medical and dental students is associated with the fact that schools of medicine and schools of dental medicine are stressful environments for the majority of students. Their under-graduate programs are often associated with significant symptoms of stress because they are among the longest and most demanding programs of study [1,72,88]. These students are exposed to stress factors that are overwhelming for some of them and could be a reason some quit their studies [89]. Thus, we see that medical and dental students report increased anxiety, frequent depression, obsessive compulsive disorders, interpersonal sensitivity, and other psychological issues/disturbances. For dental students, examinations and clinical exercises are the most frequent causes of stress [88]. Many studies found that, in undergraduate dental students, gender did not affect any of the psychological variables analyzed (distress, emotional exhaustion, stressor intensity), but study results are not consistent [82]. Many studies have compared students’ skills and previous academic outcomes with students’ cognitive traits (the perception of self-efficacy, attributions of success and failure, self-reported coping styles, dysfunctional attitudes, and irrational beliefs) [81]. Dental students also face pressure to develop clinical competencies and interpersonal skills in a short period of time and are exposed to a number of adverse factors (physical position, noise, repetitive movements, high professional competitiveness, lack of time to relax and interact with friends and family, etc.) [30]. According to literature data, for dental students, exams/testing and grades caused the majority of their academic stress and that the big-gest social stressor was having a demanding role in their personal life, such as being a wife or husband [90,91]. The study also found that higher academic stress and total stress scores were noted in those who expected to graduate with great financial debt [90,91]. Finally, it is necessary to mention that the stress resulting from medical work (examinations and clinical competencies) can have a negative impact on mental health and learning [28]. However, a structured learning program can be beneficial for students because it reduces anxiety around exams and allows students to achieve better academic results. 

## 6. Benefits of, and Perspectives on, the Further Use of Salivary Biomarkers for Research Purposes and in Practice

There are many advantages of using saliva as a biofluid for diagnostic purposes: its collection is fast, easy, non-invasive and inexpensive [92]. Its use for analysis of salivary biomarkers should be based on previous study results, which have confirmed a strong relationship between common psychological disturbances (stress, depression or anxiety) and fluctuations in levels/concentrations of stress-related salivary biomarkers (cortisol, alpha-amylase, chromogranin A and lysozyme) [54]. Results obtained from these salivary biomarkers offer insight into individuals’ stress levels, anxiety or depression, primarily using salivary cortisol, immunoglobulin A (sIgA), salivary alpha-amylase, chromogranin A, lysozyme, melatonin, and fibroblast growth factor 2 (FGF-2). Furthermore, assessment of salivary cortisol and melatonin can be helpful for differentiating between stress and depression [54]. When interpreting salivary biomarker results, however, it must be noted that sampling protocols may alter the composition of the saliva, and thus the concentration of some of the most important stress-linked analytes. It should also be taken into account that levels of stress biomarkers are changed by certain physiological and metabolic conditions (e.g., cardiovascular diseases, oral cancers, oral lichen planus, and others); thus, results are not necessarily due to stress itself.

Despite any limitations to its use, evidence/data that saliva can reflect physiological or pathological states means that salivary biomarkers can serve as diagnostic or monitoring tools in many branches of healthcare, such as medicine, dentistry, and pharmacotherapy. Additionally, data on stress-related salivary biomarkers may help our understanding of the complex relationship between psychological and neuroimmune reactions to stress and the influence of psychological stress on the organism, including during academic stress. 

In the future, more studies will likely be conducted in this area on measuring and monitoring academic and student stress with salivary biomarkers. This approach could be useful and applied in real life, for example, by helping design measures for the timely recognition/avoidance of developing psychological disturbances and (when possible) the development of psychiatric conditions/diseases. Such data could be helpful in identifying students who might need medical attention due to their perceived stress and who may develop distress-related mental health disturbances. It could also provide evidence-based support for specific public health measures for students to decrease the negative influences of emotional stress and similar disorders, which significantly affect academic performance and, later, their professional work. 

## 7. Conclusions

The organism’s complex reactions to psychological stress, such as academic stress among students, involves various endocrine and immune factors, some of which can be easily measured in saliva and used as biomarkers of stress. According to previous studies, the measurement of stress levels by salivary biomarkers (salivary cortisol, sIgA, salivary alpha-amylase, chromogranin A, lysozyme, melatonin, and others) is a simple, non-invasive and reliable way of collecting samples and obtaining data on the influence of stressful events on students. Some of these biomarkers may also by useful for measuring anxiety or depression and can even help identify students who are prone to high stress associated with exams or studying, including those at high risk for experiencing specifically negative influences of stress on their ability/capacity to study. It could also help identify the students who need greater help coping with stress in general, thus reducing their risk for the development of conditions that are influenced by stress factors (e.g., cardiovascular, autoimmune, cancer). Such data could be a basis for the potential implementation of further anti-stress measures in practice and could lead to organizing better academic courses that allow students to learn more effectively or better manage the demands of academic work. Since students are prone to stress even after graduation, these results could help diminish the impact of stress on the organism during their transition to professional life as well. More research should be done in this field on larger samples of students to gather more specific data on stress-associated salivary biomarkers. This would provide better insight into the usefulness of their interpretation in practice.

## Figures and Tables

**Figure 1 behavsci-12-00400-f001:**
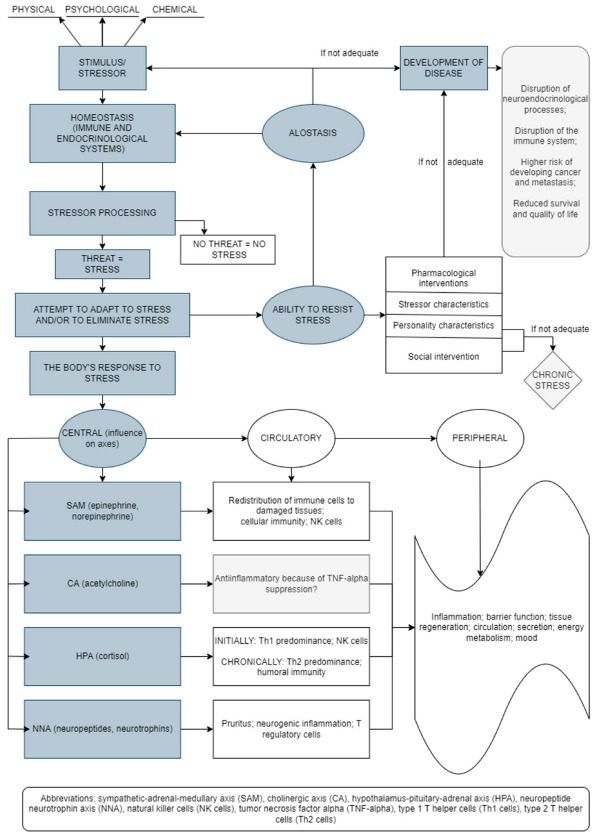
Stress-induced bodily reactions and factors involved in this network (an original scheme based on current literature data).

**Figure 2 behavsci-12-00400-f002:**
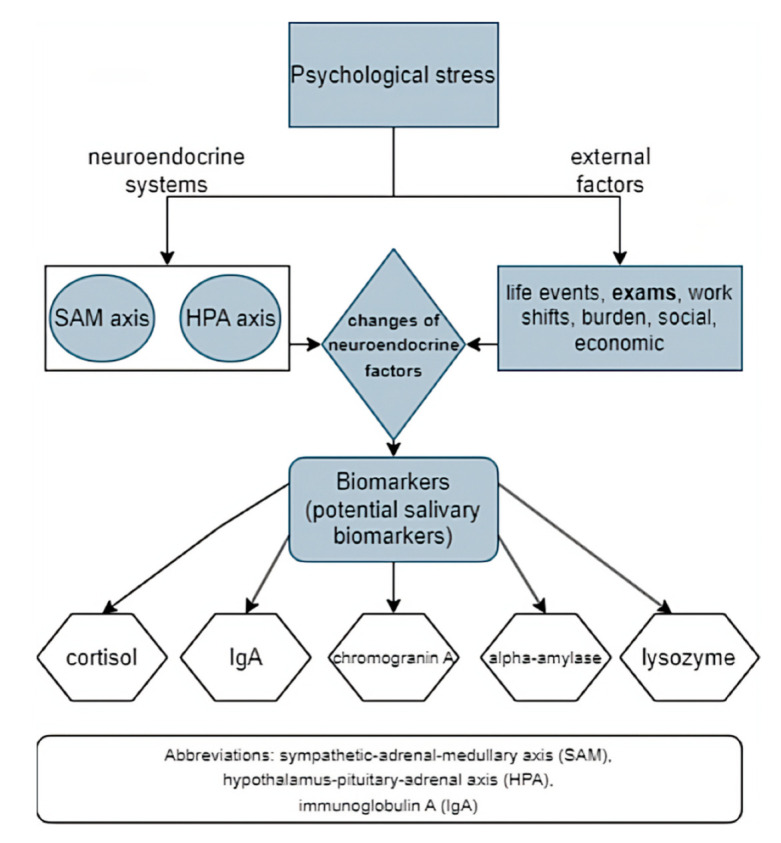
Psychological stress reaction, changes in neuroendocrine factors and potential salivary biomarkers (an original scheme based on current literature data).

**Table 1 behavsci-12-00400-t001:** Recent significant studies, involving students as subjects, on salivary cortisol and other salivary biomarkers as indicators of academic stress.

Research	Participants (N)	Analyzed Factors/Methods	Results
Bardi et al. (2011) Stress. [81]	91 college chemistry students	Stress related to examination performance in students was analyzed by salivary cortisol and dehydroepiandrosterone (DHEA) levels (before and after every major test). Students’ displacement Activities (DAs) were videorecorded during every test. Achievement was objectively recorded using grade point averages (GPA), American College Testing (ACT) scores, and final class grades.	Displacement activities under stress and salivary cortisol and DHEA levels correctly predicted whether a student would pass the semester 90% of the time. However, the method could only correctly predict who would not pass the semester 52% of the time, which is insufficient for establishing a model.
Cipra et al. (2019) PLoS One. [83]	212 medical students	Studying methods, anxiety data (STAI-T and ASSIST), and acute state anxiety (measured twice during the second term) were examined. Salivary cortisol was measured four times, and STAI-T was measured prior to the oral exams.	Surface learning approach significantly correlated with trait anxiety. Students with predominantly strategic learning methods had the most academic success with the least anxiety.
Gaab et al. (2006) Psychoneuroendocrinology. [10]	28 healthy economics students	Effects of cognitive-behavioral stress management (CBSM) in a naturalistic setting (four times a week for eight weeks) compared to a waiting control condition. Salivary cortisol measurements (awakening response and short circadian profile) were taken multiple times and on exam day along with a cognitive appraisal.	Pre-exam period: CBSM group had lower anxiety levels and fewer physical symptoms. On exam day: controls had much lower cortisol levels. Different associations between pre-exam period cortisol responses and the cognitive stress appraisal were noted (no dissociation in CBSM group). Concentrations of short circadian cortisol remained the same between groups.
Helbig et al. (2017) Physiol. Behav. [27]	46 college students	Salivary cortisol levels were measured before and after giving an oral presentation (stress condition) or while colleagues gave a presentation (control condition). A cognitive stress appraisal questionnaire and a stress and anxiety self-assessment were conducted beforehand.	Presentations led to sharp cortisol increases (as opposed to controls). Release of cortisol was not dependent on sex, but women had higher subjective feelings of stress and displayed unfavorable cognitive appraisals; men felt their own presentations posed a greater challenge than those of their colleagues.
Kamezaki et al. (2012) Psychophysiology. [28]	28 fourth-year medical students	Salivary cortisol levels and 50 circulating immune mediators were analyzed in medical students exposed to academic examination stress at seven weeks prior to, pre- and post-exam day, and one week after the exam.	Significantly increased proinflammatory and Th2 cytokines levels, as well as β-nerve growth factor linked with sharp reduction in salivary cortisol levels and feelings of anxiety when the exam ended were recorded.
Katsuura et al. (2010) Int J Psychophysiology. [37]	26 medical students	The effect of exams (a short naturalistic stressor) on cortisol levels, feelings of depression and anxiety, and values of 50 serum immune mediators (analyzed seven days prior to, immediately before, and two days after the exam).	Sharp reductions in serum MIF, MCP-3, and β-NGF levels (two days after exams), related to concomitant post-exam reductions in salivary cortisol and anxiety. Only MIF deltas were significant, which culminated when exams started (much more than pre- and post-exam period).
Lenaert et al. (2016) Psychoneuroendocrinology. [17]	90 undergraduate students (71 returned for follow-up testing)	Attentional control of students was assessed (Emotional Attentional Control Scale), and salivary cortisol was measured, before and after exams (six weeks of research, [three weeks of preparation and three weeks of exams].	Lower pre-exam self-perceived emotional attentional control levels predicted increased total diurnal cortisol secretion during stress, and a less pronounced drop post-exam day. Difficulty maintaining attention throughout prolonged stressful situations may lead to chronic HPA-axis hyperactivity.
McGregor et al. (2016) Stress. [12]	22 first-year graduate students and 30 control participants from the community	Perceived stress, lymphocyte phenotype and salivary cortisol measured with the start of classes in autumn and in the week preceding spring examinations.	Students expressed greater distress than controls in all measures (excluding basal stress level). There was a correlation between student stress status and a significant decrease in CD19+ B lymphocytes and decreased cortisol awakening response (CAR).
Ng et al. (2003) J Dent Educ. [13]	31 dental undergraduates	Students were surveyed (one-to-five rating questionnaire) before and after an hour-long written exam; their salivary cortisol, IgA and chromogranin A (CgA) were also measured.	Pre-exam, higher self-reported stress and cortisol levels were recorded. (No significant differences in IgA and CgA samples.) Pre-exam stress scores were associated with raised salivary cortisol but not IgA or CgA. Higher perceived stress levels among students generally resulted in worse exam results.
Ouda et al. (2016) J Clin Exp Pathol. [1]	90 undergraduate students	Salivary stress biomarker levels [s-cortisol, salivary alpha-amylase (sAA), and IgA] were measured in undergraduate dental students (by ELISA) by collecting two saliva samples, one before and one after the exam.	A sharp upturn of salivary stress biomarkers was noted during the assessment (stress) phase (unlike in the non-assessment [rest] phase). A strong connection was seen between salivary α-amylase and success in the academic environment, especially in males and those finishing their studies.
Pani et al. (2011) J Dent Educ. [82]	40 final-year dental students	Evaluation of stress using salivary cortisol (beginning of the semester, the final week of clinical practice, and 60 min before the final exam); adapted Displacement Activities (DAs) questionnaire.	Much higher cortisol levels were seen the final week of clinical practice (even higher during the pre-examination trial). Deviations may exist between reported feelings of stress and amount of stress exposed to during studies.
Ringeisen et al. (2019) Anxiety Stress Coping. [84]	92 students	Assessment of self-belief, risk evaluation, contextual anxiety and cortisol levels on a control day preceding an oral exam by a week, and again on exam day (anxiety was tested three times).	Continually reducing anxiety was seen after the exam (greater anxiety led to sharper declines). A correlation between anxiety and cortisol levels could not be established. Self-belief showed opposite trends to risk assessment and anxiety levels on the control day. Higher risk assessments were linked with more anxiety. A steep decrease in anxiety on exam day led to increased performance.
Schoofs et al. (2008) Stress. [29]	40 students (20 students participated in a second examination, one to four weeks later)	Measurement of cortisol and salivary alpha-amylase (sAA) on control day, as well as pre-exam and post-exam. Personality tests were also filled out.	A big increase in cortisol and sAA levels was found as a result of examination, with noticeable increases between the two measurements. No differences between the sexes or for oral contraceptive use were found.
Suh M. (2018) J Neurosci Nurs. [74]	36 female college students	Salivary cortisol measured six times a day during a phase without exams and during a stressful phase; specificities in sleep quality and anxiety were measured by actigraph and STAI (Spielberger State-Trait Anxiety Inventory).	Students with heightened feelings of anxiety enjoyed better quality of sleep; subjects experiencing poor sleep had heightened cortisol secretion during waking hours. Subjects more prone to anxiety had greater cortisol levels during exams.
Viena et al. (2012) Biol Psychol. [71]	30 undergraduate students	Effects of stress on cortisol and s-IgA responses were analyzed through/by two phases of research (four weeks before exams and during final exams). Students were tested between four and six PM (when cortisol secretion is low/decreasing [not at the circadian nadir]). Students completed the Perceived Stress Scale-10 (PSS-10) and other questionnaires during both study phases.	Acute stressors strongly increase salivary cortisol levels during stress-free phases, but not when a mild chronic stress is ongoing, where s-IgA levels showed an increase on both occasions. During continuous mild stress, perceived stress is reflected in a heightened cortisol response.

Abbreviations: ASSIST—the approaches-and-study-skills-inventory-for-students; DAs—displacement activities; GPA—grade point average; ACT—American college testing (ACT); STAI-T—state-trait-anxiety inventory; ASSIST—the approaches-and-study-skills-inventory-for-students; CBSM—cognitive behavioral stress management; MIF—macrophage migration inhibitory factor (MIF); MCP-3—monocyte chemoattractant protein; β-NGF—β-nerve growth factor; CAR—cortisol awakening response; ELISA - Enzyme-linked Immunosorbent Assay; PSS-10—perceived stress scale-10.

## Data Availability

Not applicable.

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
