# Peer review of "A Review of Psychological Stress among Students and Its Assessment Using Salivary Biomarkers"

_behavsci, 2022, doi:10.3390/bs12100400_

Round 1

Reviewer 1 Report

The manuscript describe the potential use of salivary biomarkers to  assess stress among students. I have some suggestions to improve the manuscript:

1. It is well-know that the sampling protocols may alterate the composition of saliva and then the concentration of some of the most important analytes linked to stress (as example see: https://dx.doi.org/10.1016/j.microc.2017.04.033, https://dx.doi.org/10.1016/j.microc.2017.02.032). Thus i strongly suggest to discuss this topic in the review aimed at providing a clear overview of the salivary analysis at readers.

2. I suggest to discuss that potential stress biomarkers are in related to other physiological/metabolic conditions, thus are no unique for the stress itself. 

3. Could you comment on the future perspective of the saliva analysis to monitor stress among students? Do you believe that this approach can be applied in real scenario? If yes, could you provide an example? What will be the main advantage of using this approach in real scenario?

Author Response

Dear Editor and Reviewers,

thank You for Your valuable time, effort and useful contribution You have put into assesing our previous version of the manuscript entitled „Psychological stress among students and possibilities for its assessment by salivary biomarkers: a narrative review”.  We appreciate the input You have given because it has definitely improved our manuscript. Also, we engaged a professional to lecture the writing.

We have given careful consideration to each comment as follows:

REVIEWER:  1 - Comments and Suggestions for Authors

The manuscript describe the potential use of salivary biomarkers to  assess stress among students. I have some suggestions to improve the manuscript:

  1. It is well-know that the sampling protocols may alterate the composition of saliva and then the concentration of some of the most important analytes linked to stress (as example see:

https://dx.doi.org/10.1016/j.microc.2017.04.033,   https://dx.doi.org/10.1016/j.microc.2017.02.032). Thus i strongly suggest to discuss this topic in the review aimed at providing a clear overview of the salivary analysis at readers.

-Thank You! We corrected the text and added more data concerning the influence of sampling protocols on the results of salivary biomarkers and concerning possible alteration of the composition of saliva and then the concentration of some of the most important analytes linked to stress. We also added additional references important for this subject.

  1. I suggest to discuss that potential stress biomarkers are related to other physiological and metabolic conditions, thus are no unique for the stress itself. 

-Thank You! We added more data on the relation between potential stress biomarkers and other physiological and metabolic conditions. Also, we added some new references in this field.

  1. Could you comment on the future perspective of the saliva analysis to monitor stress among students? Do you believe that this approach can be applied in real scenario? If yes, could you provide an example? What will be the main advantage of using this approach in real scenario?

-Thank You! We corrected the text and added additional comments with various information and data on this subject, including some examples. In new version of the manuscript, it is written in the final part of the manuscript.

Reviewer 2 Report

SUMMARY AND GENERAL CONSIDERATIONS

This well-structured work aims to show current data on salivary biomarkers for physiological stress among graduate students. It generates awareness of a matter of interest for society, and can inspire the scientific community to, for instance, develop efficient methods and technologies for the fast monitoring of such biomarkers.

They first present the general physiologic particularities of stress (without obviating gender differences) and the stress-related inflammatory reaction, which are indeed related to the stress salivary biomarkers presented afterwards. At the end, they define academic stress and describe diverse stress factors and how they affects students, considering diverse study fields and gender. Conclusions efficiently sum up the main points and bring home information brought up during the whole article.

DETAILED COMMENTS

Line 69-17: Why in parenthesis and between stops at the same time?

Figure 1: Very useful flow chart. I would suggest the authors to colour (at least with a grey scale) the diverse boxes considering level/nature of its content. That would ease comprehension of such a useful figure.

130: Unnecessary space in front of ref 19.

Table 1: A landscape Table might be more easily readable. Please make sure that the Table caption stays close and together in top of the table (maybe by inserting it in a first table cell).

Author Response

Dear Editor and Reviewers,

thank You for Your valuable time, effort and useful contribution You have put into assesing our previous version of the manuscript entitled „Psychological stress among students and possibilities for its assessment by salivary biomarkers: a narrative review”.  We appreciate the input You have given because it has definitely improved our manuscript. Also, we engaged a professional to lecture the writing.

We have given careful consideration to each comment as follows:

REVIEWER:  2

This well-structured work aims to show current data on salivary biomarkers for physiological stress among graduate students. It generates awareness of a matter of interest for society, and can inspire the scientific community to, for instance, develop efficient methods and technologies for the fast monitoring of such biomarkers. They first present the general physiologic particularities of stress (without obviating gender differences) and the stress-related inflammatory reaction, which are indeed related to the stress salivary biomarkers presented afterwards. At the end, they define academic stress and describe diverse stress factors and how they affects students, considering diverse study fields and gender. Conclusions efficiently sum up the main points and bring home information brought up during the whole article.

DETAILED COMMENTS

Line 69-17: Why (text) in parenthesis and between stops at the same time?

-Thank You! We corrected it and removed them.

Figure 1: Very useful flow chart.  I would suggest the authors to colour (at least with a grey scale) the diverse boxes considering level/nature of its content. That would ease comprehension of such a useful figure.

-Thank You! We corrected Figure 1 and added the colour (grey) to it.

130: Unnecessary space in front of ref 19.

-Thank You! We removed it.

Table 1: A landscape Table might be more easily readable. Please make sure that

-Thank You! We changed this form of the pages and now there is a landscape.

The Table caption stays close and together in top of the table (maybe by inserting it in a first table cell).

-Thank You! We changed the format of the title of  Table 1 according to Your suggestions.

Reviewer 3 Report

Dear Authors,

Thank you for submitting your paper to Behavioral Sciences. The topic is interesting and I hope that my remarks and suggestions will be useful in order to improve the quality of the manuscript.

The paper has some formatting deficiencies like:

Line 2 – you should mention what type of paper is it;

Line 6 – number corresponding to the affiliation should be placed after the name of the author and not in front of it;

Line 14 – The corresponding author is not properly written in MDPI style. It should be „Correspondence: e-mail and telephone of the author”

Line 55 – You should briefly mention how did you select the literature for this narrative review. What did you include and what did you exclude.

Line 58 – The pieces of information in Fig. 1 requires citation.

Line 142 – Section 3: please correct „IL-b” with either IL-1 beta or IL-1b

Line 245 – Please mention in the section a few methods to assess the level of academic stress.

Table 1 – I don’t think that the capitals are needed.

Line 320 – Conclusions section should be reorganised in such a way that it will emphasise in a synthetic manner a clear correlation between the salivary biomarker mentioned in the paper and the level of academic stress. Please highlight the practical perspectives that your review open. Also a „take-home” message is needed.

Line 343- Authors contribution should be written in the MDPI style. The way you did it doesn’t respect the authorship criteria. Please consult the recommendation for authors.

Line 347: Funding should be followed by „:” and afterwards, on the same line the information.

Line 350: the same remark.

I would suggest to add one more figure in a much easy to understand way and in a more friendly design.

English needs improvements so please make sure that a native checks the manuscript.

Kind regards!

Author Response

Dear Editor and Reviewers,

thank You for Your valuable time, effort and useful contribution You have put into assesing our previous version of the manuscript entitled „Psychological stress among students and possibilities for its assessment by salivary biomarkers: a narrative review”.  We appreciate the input You have given because it has definitely improved our manuscript. Also, we engaged a professional to lecture the writing.

We have given careful consideration to each comment as follows:

REVIEWER: 3

 The paper has some formatting deficiencies like:

 Line 2 – you should mention what type of paper is it;

-We added more information to this line 2.

Line 6 – number corresponding to the affiliation should be placed after the name of the author and not in front of it;

-Thank You! We corrected the specific places of these numbers and moved the numbers to correct positions.

Line 14 – The corresponding author is not properly written in MDPI style. It should be „Correspondence: e-mail and telephone of the author”

-Thank You! We corrected it.

Line 55 – You should briefly mention how did you select the literature for this narrative review. What did you include and what did you exclude.

-Thank You! We added more information on the selection the literature which was a basis for the manuscript.

Line 58 – The pieces of information in Fig. 1 requires citation.

-Thank You! We corrected it and added the numbers of the references which were sources and a basis for the figure.

Line 142 – Section 3: please correct „IL-b” with either IL-1 beta or IL-1b

-Thank You! We corrected it into IL-1b.

Line 245 – Please mention in the section a few methods to assess the level of academic stress.

-Thank You! We added more information on methods which are generally used for an assesment of psychological stress.

Table 1 – I don’t think that the capitals are needed.

-Thank You! We removed them.

Line 320 – Conclusions section should be reorganised in such a way that it will emphasise in a synthetic manner a clear correlation between the salivary biomarker mentioned in the paper and the level of academic stress. Please highlight the practical perspectives that your review open. Also a „take-home” message is needed.

-Thank You! We corrected the Conclusion and, according to Your suggestions, we added more specific information on this place.

Line 343- Authors contribution should be written in the MDPI style. The way you did it doesn’t respect the authorship criteria. Please consult the recommendation for authors.

-Thank You! We corrected this style according to journal style.

Line 347: Funding should be followed by „:” and afterwards, on the same line the information.

Line 350: the same remark.

-Thank You! We corrected it and wrote it according to the journal rules.

I would suggest to add one more figure in a much easy to understand way and in a more friendly design.

-Thank You! We added one additional figure (Figure 2).

English needs improvements so please make sure that a native checks the manuscript.

              -Thank You! We corrected it – we asked a native speaker and the text is checked.

Best regards,

Authors

Round 2

Reviewer 1 Report

Dear Authors, all questions were discussed and the manuscript modified accordingly. So I suggest to accept it in Behaviour Science.

Author Response

Dear Reviewer,

thank You again for Your valuable time, effort and useful contribution You have put into assessing our previous version of the manuscript entitled „Psychological stress among students and possibilities for its assessment by salivary biomarkers: a narrative review”.  Also, we engaged a native speaker to lecture the writing.

                   RESPONSE

REVIEWER 1.

Dear Authors, all questions were discussed and the manuscript modified accordingly. So I suggest to accept it in Behaviour Science.

-Thank You! We checked the text and English in the manuscript by a native speaker.

Best regards,

Authors

Reviewer 3 Report

Dear authors,

Thank you for resubmitting the revised version.

I hope my remarks will be useful in order to increase the quality of the manuscript.

1) The manuscript that I have downloaded doesn't contain the layout of the journal;

2) I cannot see the numbers of the lines;

3) The title must be rephrased in order to reflect the review type of article;

4) Page 2: In the highlighted paragraph: keep in mind that all articles included in international data bases are written in English;

5) Page 3 - I Fig. 1 original or does it require copyrights?

6) Correct the spelling of "IL-1 Beta" using the special symbol for "Beta";

7) Conclusions section is not synthetic enough. please rephrase.

Best regards!

Author Response

Dear Reviewer,

thank You again for Your valuable time, effort and useful contribution You have put into assessing our previous version of the manuscript entitled „Psychological stress among students and possibilities for its assessment by salivary biomarkers: a narrative review”.  Also, we engaged a native speaker to lecture the writing.

We have given careful consideration to each comment as follows:

REVIEWER 3.

1) The manuscript that I have downloaded doesn't contain the layout of the journal;

-Thank You! We corrected the outlook of the manuscript and changed it according to journal rules. We hope that it is visible to you.

2) I cannot see the numbers of the lines;

-Thank You! We corrected outlook of the manuscript  and added the numbers of the lines - we hope that they are visible to you.

3) The title must be rephrased in order to reflect the review type of article

-Thank You! We corrected the title and rephrase it according to your recommandations.

4) Page 2: In the highlighted paragraph: keep in mind that all articles included in international data bases are written in English;

-Thank You! We corrected this part of the text and also removed some incorrect or unnecessary words concerning English..

5) Page 3 - I Fig. 1 original or does it require copyrights?

-Thank You! We added an information that both schemes (figures) are original .

6) Correct the spelling of "IL-1 Beta" using the special symbol for "Beta";-ja

-Thank You! We corrected this word and added specific symbol, as you said.

7) Conclusions section is not synthetic enough. please rephrase.

-Thank You! We rephrased the text of the Conclusion to be more synthetic.

+ Check english according to journal rules

-Thank You! We engaged the native speaker who checked and corrected the text.

Best regards,

Authors

Round 3

Reviewer 3 Report

I would suggest the authors to rephrase the title. From my point of view is not proper.

Author Response

Dear Reviewer, thank You again!

According to your suggestion, we rephrased the title - with help of a native speaker who checked it.

We hope that you consider that it is correct now.

Best regards,

Authors